# The Serbian version of the Pandemic-Related Pregnancy Stress Scale (PREPS-SRB)–A validation study

**Konstantin Kostić**[1,2]*, **Aleksandra Kostić**[3], **Aleksandra Petrović**[1,2], **Andrija Vasilijević**[2], **Jelena Milin-Lazović**[4]

**1** Clinic for Gynecology and Obstetrics Narodni Front, Belgrade, Serbia, **2** Faculty of Medicine, University of Belgrade, Belgrade, Serbia, **3** Community Health Center "Dr Simo Milošević", Belgrade, Serbia, **4** Institute for Medical Statistics and Informatics, Faculty of Medicine, University of Belgrade, Belgrade, Serbia

* k.ginekos@gmail.com

**Data Availability Statement:** All relevant data are within the paper and its Supporting Information files.

## Abstract

Pregnancy is a sensitive period in a woman's life when psychological distress can have negative consequences for the mother and fetus. Prolonged and intensified symptoms of anxiety and depression caused by the COVID-19 pandemic increase the risk of maternal and fetal health complications. The Pandemic-Related Pregnancy Stress Scale (PREPS) is a thoroughly designed tool that helps determine and analyze stress among pregnant women during pandemics in three domains: Preparedness in childbirth, (2) Infection, and (3) Positive Appraisal. A cross-sectional study included 189 pregnant women attending a community health center, "Dr Simo Milošević," in Belgrade, Serbia, from January to February 2022. Pregnant women anonymously completed a questionnaire as part of the study. The mean scores for those three domains are as follows: Preparedness (2.4 ± 0.9), Infection stress (2.8 ± 1.1), and Positive Appraisal (3.7 ± 0.9). Internal consistency of the PREPS questionnaire for PREPS-Total (α = 0.867). An explanatory factor analysis of the PREPS showed that the Serbian version of the Pandemic-Related Pregnancy Stress Scale has good psychometric properties. The Kaiser-Meyer-Olkin Measure of Sampling Adequacy (KMO) was found to be 0.860, indicating a high degree of sampling adequacy. Additionally, Bartlett's Test of Sphericity yielded a statistically significant result ($\chi^2$ = 1564.206, df = 105, p < 0.001). The CFA showed very good fit indices for the Serbian sample, confirming the factor structure of the original English version. The RMSEA value of 0.056 (0.036–0.075) and values for fit indices TLI (0.961) and CFI (0.974) were above the cut-off of ≥0.95, indicating an excellent fit. All standardized factor loadings were statistically significant and ranged from 0.50 to 0.85. The PREPS-SRB questionnaire serves as a valuable tool for Serbian healthcare professionals, allowing them to identify pregnant women experiencing significant stress related to the COVID-19 pandemic.

**Funding:** The author(s) received no specific funding for this work.

**Competing interests:** The authors have declared that no competing interests exist.

## Introduction

Pregnancy is a sensitive period in a woman's life when psychological distress can have negative consequences for the mother and fetus. Many studies have shown that prolonged and intensified symptoms of anxiety and depression increase the risk of miscarriage, premature birth, low APGAR score, postpartum depression, and increased incidence of prenatal infections [1–3]. Mothers who have experienced high levels of stress during pregnancy give birth to children who are at increased risk of developing mental illnesses later in life [4, 5]. Additionally, Sabbagh et al. showed in their retrospective cohort study that stressful life events during pregnancy, along with an family history of congenital disabilities, may be the risk factor for the development of non-syndromic orofacial clefts [6]. One cross-sectional study in the United States performed by Weiner et al. showed that newborns who were raised by mothers who were under psychological distress during the COVID-19 pandemic showed a median reduction in white matter and left amygdala volumes compared with infants of mothers with low-stress levels [7].

The COVID-19 pandemic caused significant disruptions to healthcare access. Prenatal appointments were limited or conducted virtually, reducing opportunities for in-person interaction with healthcare providers. Social distancing measures also limited access to traditional support systems from family and friends, further isolating expectant mothers. These disruptions to routine and social connection significantly amplified stress levels in pregnant women, potentially increasing the risk for anxiety, depression, and other mental health challenges [8, 9].

Pregnant women and women in labor are at increased risk of developing more severe clinical symptoms of COVID-19 infection compared to non-pregnant women. Pregnant women with COVID-19 infection are at increased risk of stillbirth and preterm birth and may be at increased risk of other complications during pregnancy [10]. A cohort study by Wilkinson et al. indicates that COVID-19 infection in pregnant women has no consequences for fetal growth, maternal, and neonatal outcomes compared to women not infected with COVID-19 [11]. On the other hand, a retrospective study performed by Sero et al. showed there was a statistically significantly higher rate of pre-eclampsia, abnormal non-stress test, perinatal asphyxia, and birth weight in women who lost their lives during the COVID-19 pandemic in comparison to pregnant women who survived COVID-19 in good health [12]. This initial concern and rapidly evolving information about the virus and its transmission created a climate of fear and uncertainty [13].

Since the beginning of the COVID-19 pandemic, but before conducting this study in January of 2022, only two scales were initialy found which were specifically designed for measuring stress in pregnant women during the COVID-19 pandemic: The Pandemic-Related Pregnancy Stress Scale (PREPS) [14] and The COVID-19 Perinatal Perception Questionnaire (COVID19-PPQ) [15]. In this study, we chose PREPS over COVID19-PPQ because of better psychometric properties and because it was initially conducted on a larger sample.

So far, validation of the PREPS questionnaire has been done in several countries, including USA [14], Spain [16], Germany [17], Poland [18], Greece [19], Korea [20] and Italy [21]. However, the PREPS questionnaire hasn't been validated in the Balkan region. The appropriate translation of PREPS for the Serbian-speaking population will allow it to be potentially used in future research to assess pregnancy-related stress due to the pandemic and to guide early intervention strategies.

According to the Ministry of Health of Serbia, until January of 2022, there were more than 1.4 million confirmed cases of COVID-19, around 2400 hospitalized patients and around 13,000 COVID-19 death cases, in the population of 7.6 million Serbian residents [22].

Worldwide, until January of 2022, there was around 330 million confirmed cases of COVID-19 predominately in Americas and Europe [23] and around 5.5 COVID-19 death cases [24].

This study aimed to validate PREPS in the Serbian-speaking population of pregnant women.

## Material and methods

A cross-sectional study included pregnant women attending a community health center, "Dr Simo Milošević," in Belgrade, Serbia, from 17th of January to 28th of February 2022. Pregnant women anonymously completed a written questionnaire during the outpatient appointments as part of the study, which was approved by the community health center's ethical committee (Ethics approval letter No. 28719 & 126/1).

Participation in the study was voluntary. All participants provided written informed consent before data collection.

### Questionnaires

The questionnaire used in this study was comprehensive, gathering information across four areas. The first section focused on sociodemographic details such as age, education level, financial and marital status. Obstetric factors were then explored, including gestational age, pregnancy planning, parity, any high-risk factors, chronic diseases, and infertility treatment history. The COVID-19 history section delved into participants' experiences with the virus, including prior infection before or during pregnancy, their vaccination status, job loss due to the pandemic, and any loss of loved ones from COVID-19. Finally, the Pandemic-Related Pregnancy Stress Scale (PREPS) questionnaire [14] was used as thoroughly designed toot that helps assess the unique stress of pregnant women during a global health crisis. It was created by Heidi Preis et al. and their study was initially conducted in 2020 on pregnant women across the US via social media. An exploratory and confirmatory factor analysis of this original version provided insight into three domains: (1) Preparedness in childbirth– 7 questions, (2) Infection– 5 questions, and (3) Positive Appraisal– 3 questions, and showed acceptable to good reliability for each domain (Cronbach's α: 0.86, 0,81 and 0,68, respectively) and excellent fit (CFI = 0.93, TLI = 0.91, RMSEA 0.07). For every statement, the responses ranged from "Very Little" (1), "Little" (2), "Some" (3), "Much" (4), "Very Much" (5). For each domain, mean was calculated.

The original authors gave permission to use PREPS.

We initially translated the PREPS from English to Serbian (forward translation) using two independent translators who were fluent in English. These two versions were then compared and analysed. A third professional translator, unaware of the original English version, performed a back translation from Serbian to English. Subsequently, all translators reviewed the three translations together to identify and resolve any discrepancies. Following modifications and consensus among the translators, we developed the final Serbian version of the PREPS. This version was then tested on ten pregnant women to assess comprehension and gather feedback on clarity, and no significant issues were reported during this process.

### Statistical analysis

Data analysis employed a combination of descriptive and analytical statistics. Descriptive statistics included measures of central tendency (average and median), variability (standard deviation and range), and absolute and relative frequencies.

Since the PREPS questionnaire consists of 15 items, the minimum suggested sample size for this study was 75 participants [25]. Internal consistency analyses of the PREPS questionnaire

were performed using Cronbach's alpha coefficients. Values above 0.70 were considered satis-factory. Construct validity was examined using explanatory factor analysis. Factor extraction was performed using the Principal Components Analysis method with Varimax rotation. To assess the suitability of the data for factor analysis, the Kaiser–Meyer–Olkin (KMO) measure and Bartlett's test of sphericity were calculated. The construct validity of the Serbian version of the PREPS was tested using confirmatory factor analysis (CFA). In addition, the Comparative-Fit Index (CFI), Tucker–Lewis index (TLI), and the Root Mean Square Error of Approxima-tion (RMSEA) were used for model fit. Values of CFI and TLI above 0.90 were considered ade-quate, whereas RMSEA value below 0.06 indicated an acceptable model fit. All tests were two-tailed. In all analyses, the significance level was set at 0.05. Statistical analysis was done using Amos 21 (IBM SPSS Inc., Chicago, IL, USA, 2012) and IBM SPSS Statistics 25 software.

## Results

A total of 189 participants enrolled in this study. The sociodemographic characteristics, obstet-ric and COVID-19 history of the participants are presented in Table 1.

Regarding pregnancy planning, 91.5% of respondents indicated that their current preg-nancy was planned, while 8.5% reported it was unplanned. In terms of previous deliveries, 55% of respondents had previous deliveries, while 45% were primiparous. Among those who had deliveries before, the distribution was as follows: 73 respondents had one delivery, 28 had two deliveries, 2 had three deliveries, and 1 had four deliveries. Concerning the current preg-nancy's risk status, 14.3% of respondents considered their pregnancy high-risk, while 67.2% did not, and 18.5% were unsure. 11.1% of respondents reported suffering from chronic ill-nesses. Regarding infertility treatment, 6.3% of respondents and/or their partners had received treatment. Regarding the duration of attempts to conceive, 14.3% had been trying for more than a year, while 85.7% had been trying for up to a year. Finally, regarding emotional or psy-chiatric issues, 3.7% of respondents reported currently experiencing such issues, while 96.3% did not.

During their current pregnancy, 6 out of 189 participants (3.2%) reported being in contact with a COVID-positive person, while 183 participants (96.8%) reported no such contact. Moreover, 68 out of 189 participants (36%) reported being in contact with a COVID-positive person before their current pregnancy, while 121 participants (64%) reported no such contact. Regarding COVID-19 infections, 78 participants (41.3%) had a COVID-19 infection before their current pregnancy, of which 85 participants (45%) reported no prior infection, and 26 participants (13.8%) were unsure. During their current pregnancy, 40 participants (21.2%) reported having a COVID-19 infection, with 131 participants (69.3%) reporting no infection, and 18 participants (9.5%) being unsure. Regarding vaccination, 68 participants (36%) reported being vaccinated against COVID-19, while 121 participants (64%) reported not being vaccinated. Among those vaccinated, 1 participant (0.5%) received one dose, 51 participants (27%) received two doses, and 16 participants (8.5%) received three doses. Additionally, 138 participants (73%) reported that someone in their household had tested positive for COVID-19, while 51 (27%) reported no such occurrence. Finally, 24 participants (12.7%) reported the death of someone close due to a COVID-19 infection, while 165 participants (87.3%) reported no such occurrence.

The mean scores for the three domains assessed using PREPS are presented in Table 2.

The Kaiser-Meyer-Olkin Measure of Sampling Adequacy (KMO) was found to be 0.860, indicating a high degree of sampling adequacy. Additionally, Bartlett's Test of Sphericity yielded a statistically significant result ($\chi^2$ = 1564.206, df = 105, p < 0.001).

**Table 1. Sociodemographic characteristics, obstetric and COVID-19 history of participants.**

| Sociodemographic characteristics | Value |
|---|:---:|
| Age (years) | 31.5 ± 5.2 |
| Current gestational age (gestational weeks) | 23.2 ± 10.2 |
| Financial status assessment–N (%) | |
| No response | 11 (5.8) |
| Above average | 14 (7.4) |
| Below average | 13 (6.9) |
| Average | 151 (79.9) |
| Marital status–N (%) | |
| Cohabiting | 55 (29.1) |
| Unmarried | 19 (10.1) |
| Divorced | 3 (1.6) |
| Married | 112 (59.3) |
| Current employment status–N (%) | |
| Employed full-time | 142 (75.1) |
| Fixed-term contract employee | 25 (13.2) |
| Working but unregistered | 2 (1.1) |
| Unemployed | 18 (9.5) |
| Student | 1 (0.5) |
| Student, employed part-time | 1 (0.5) |
| Education–N (%) | |
| Primary school | 3 (1.6) |
| High school | 66 (34.9) |
| College | 34 (18) |
| University | 86 (45.5) |
| Current residence–N (%) | |
| Urban area | 171 (90.5) |
| Rural area | 18 (9.5) |
| Chronic illnesses–N (%) | |
| Yes | 21 (11.1) |
| No | 127 (88.9) |
| Emotional or psychiatric issues–N (%) | |
| Yes | 7 (3.7) |
| No | 182 (96.3) |
| **Obstetric history** | **Value** |
| Planned pregnancy–N (%) | |
| Yes | 173 (91.5) |
| No | 16 (8.5) |
| Previous deliveries–N (%) | |
| Yes | 104 (55) |
| No | 85 (45) |
| Current pregnancy considered high-risk–N (%) | |
| Yes | 27 (14.3) |
| No | 127 (67.2) |
| Not Sure | 35 (18.5) |
| Infertility treatment for one/both partners–N (%) | |
| Yes | 12 (6.3) |
| No | 177 (93.7) |

(*Continued*)

**Table 1.** (Continued)

| Sociodemographic characteristics | Value |
|---|---|
| Duration of attempts to conceive–N (%) | |
| More than a year | 27 (14.3) |
| Up to a year | 162 (85.7) |
| **COVID-19 history** | **Value** |
| COVID-19 vaccination status–N (%) | |
| Vaccinated | 68 (36) |
| Not Vaccinated | 121 (64) |
| COVID-19 infection before current pregnancy–N (%) | |
| Yes, home treatment | 78 (41.2) |
| No | 85 (45) |
| Not Sure | 26 (13.8) |
| COVID-19 infection during current pregnancy–N (%) | |
| Yes, home treatment | 40 (21.2) |
| No | 131 (69.3) |
| Not Sure | 18 (9.5) |
| COVID-positive individuals from the same household–N (%) | |
| Yes | 138 (73) |
| No | 51 (27) |
| The loss of a loved one from COVID-19 infection–N (%) | |
| Yes | 24 (12.7) |
| No | 165 (87.3) |

Internal consistency of the PREPS questionnaire for PREPS-Total (α = 0.867), PREPS-"Preparedness Stress" domain (α = 0.805), and PREPS-"Infection Stress" (α = 0.865) showed good scores, and for PREPS-"Positive Appraisal" (α = 0.765) showed adequate score.

Regarding the total variance explained by the principal components, the initial eigenvalues revealed that the first three components accounted for 67% of the total variance. The extraction sums of squared loadings and rotation sums of squared loadings further supported this, with the first three components explaining 67.005% and 52.435% of the cumulative variance, respectively.

In the rotated component matrix, three components were identified (Table 2). The first component showed high loadings for items related to concerns about adequate preparation for childbirth, potential disruptions to childbirth plans, and fears of separation from the baby after delivery due to the COVID-19 pandemic (items 1–7). The second component included items concerning COVID-19 infection risks during pregnancy and childbirth, both personally and for the baby (items 8–12). The third component consisted of items related to the positive psychological aspects of pregnancy, such as finding strength and value in pregnancy and parenthood (items 12–15).

The three-factor structure of the PREPS questionnaire has been validated, and the results are presented in Tables 3 and 4. The CFA showed very good fit indices for the Serbian sample, confirming the factor structure of the original English version. The RMSEA value of 0.056 (0.036–0.075) was below the suggested value of ≤0.08. Values for fit indices TLI (0.961) and CFI (0.974) were above the cut-off of ≥0.95, indicating an excellent fit. All standardized factor loadings were statistically significant and ranged from 0.50 to 0.85 (Fig 1).

**Table 2. PREPS-SRB domain means and standardized factor loadings for principal component analysis.**

| | | | Means ± SD | Component | | | Cronbach's α |
|---|---|---|---|---|---|---|---|
| | | | | 1 | 2 | 3 | |
| **Pandemic-Related Pregnancy Stress Scale (PREPS) domains** | **PREPAREDNESS** | 1. I am worried I will not be prepared for the birth due to the pandemic restrictions | 2.4 ± 0.9 | **0.78** | 0.214 | -0.017 | 0.805 |
| | | 2. I am worried that the pandemic could ruin my birth plans | | **0.72** | 0.353 | 0 | |
| | | 3. I am worried I will not be able to have someone with me during the delivery | | **0.556** | 0.245 | 0.137 | |
| | | 4. I am concerned about being separated from my baby after the delivery because of the pandemic | | **0.679** | 0.233 | 0.22 | |
| | | 5. I am concerned that I won't get the prenatal care I need because of COVID-19 | | **0.799** | 0.251 | -0.148 | |
| | | 6. I am concerned that people won't be able to help me care for my baby after birth | | **0.833** | 0.236 | -0.119 | |
| | | 7. I am concerned that I am not getting enough healthy food or sleep or exercise because of COVID-19 restrictions | | **0.654** | 0.268 | -0.093 | |
| | **INFECTION** | 8. I am concerned about going to prenatal care appointments due to COVID-19 | 2.8 ± 1.1 | 0.38 | **0.627** | 0.016 | 0.865 |
| | | 9. I am concerned that a COVID-19 infection could harm my pregnancy (such as miscarriage or preterm birth) | | 0.476 | **0.664** | -0.011 | |
| | | 10. I am worried that I might get COVID-19 when I go to the hospital to deliver | | 0.278 | **0.88** | 0.073 | |
| | | 11. I am worried that my baby could get COVID-19 at the hospital after birth | | 0.268 | **0.863** | 0.154 | |
| | | 12. I am concerned that a COVID-19 infection could harm my baby | | 0.285 | **0.837** | 0.103 | |
| | **POSITIVE APPRAISAL** | 13. I feel that being pregnant is giving me strength during the pandemic | 3.7 ± 0.9 | -0.112 | 0.021 | **0.835** | 0.765 |
| | | 14. I feel that COVID-19 is helping me appreciate my pregnancy more | | 0.128 | 0.219 | **0.724** | |
| | | 15. I think about having a baby to help me get through the pandemic hardships | | -0.054 | -0.018 | **0.902** | |

SD—standard deviation

## Discussion

The mean scores and internal consistency scores for the three domains assessed using the PREPS are as follows: Preparedness (2.4 ± 0.9; α = 0.805), Infection stress (2.8 ± 1.1; α = 0.865), and Positive Appraisal (3.7 ± 0.9; α = 0.865). Internal consistency of PREPS-Total was α 0.867.

To assess the psychometric properties of the PREPS, an explanatory factor analysis was conducted. This analysis revealed a factorial structure comprising three distinct factors: Factor 1,

**Table 3. Confirmatory factor analysis for the Serbian version of PREPS.**

| Chi-Squared Goodness of Fit | df | RMSEA (90% CI) | CFI | TLI |
|---|---|---|---|---|
| 111.957 | 70 | 0.056 (0.036–0.075) | 0.974 | 0.961 |

df—degrees of freedom RMSEA—root mean square error of approximation, CFI—comparative fit index, TLI—Tucker-Lewis index

**Table 4. Comparative analysis of Cronbach's α indices and psychometric properties of the PREPS in different studies [14, 16–21].**

| | Cronbach's α | | | | Confirmatory factor analysis (CFA) | | |
|---|---|---|---|---|---|---|---|
| | PREPS "Preparedness Stress" | PREPS "Infection stress" | PREPS "Positive Appraisal" | PREPS Total | Tucker-Lewis index (TLI) | Comparative Fit Index (CFI) | Root Mean Square Error of Approximation (RMSEA) |
| Serbia | 0.805 | 0.865 | 0.765 | 0.867 | 0.961 | 0.974 | 0.056 |
| USA | 0.81 | 0.86 | 0.68 | /no data/ | 0.91 | 0.93 | 0.07 |
| Germany | 0.81 | 0.86 | 0.71 | /no data/ | 0.902 | 0.920 | 0.073 |
| Spain | 0.65 | 0.60 | 0.55 | 0.77 | /do data/ | 0.90 | 0.05 |
| Italy | 0.76 | 0.857 | 0.747 | /no data/ | /no data/ | /no data/ | /no data/ |
| Poland (3-factor model) | 0.824 | 0.882 | 0.691 | 0.858 | 0.974 | 0.979 | 0.054 |
| Greece (3-factor model) | 0.77 | 0.83 | 0.88 | 0.85 | 0.880 | 0.897 | 0.124 |
| Korea | 0.81 | 0.85 | /no data/ | 0.87 | /no data/ | /no data/ | 0.06 |

PREPS = The Pandemic-Related Pregnancy Stress Scale

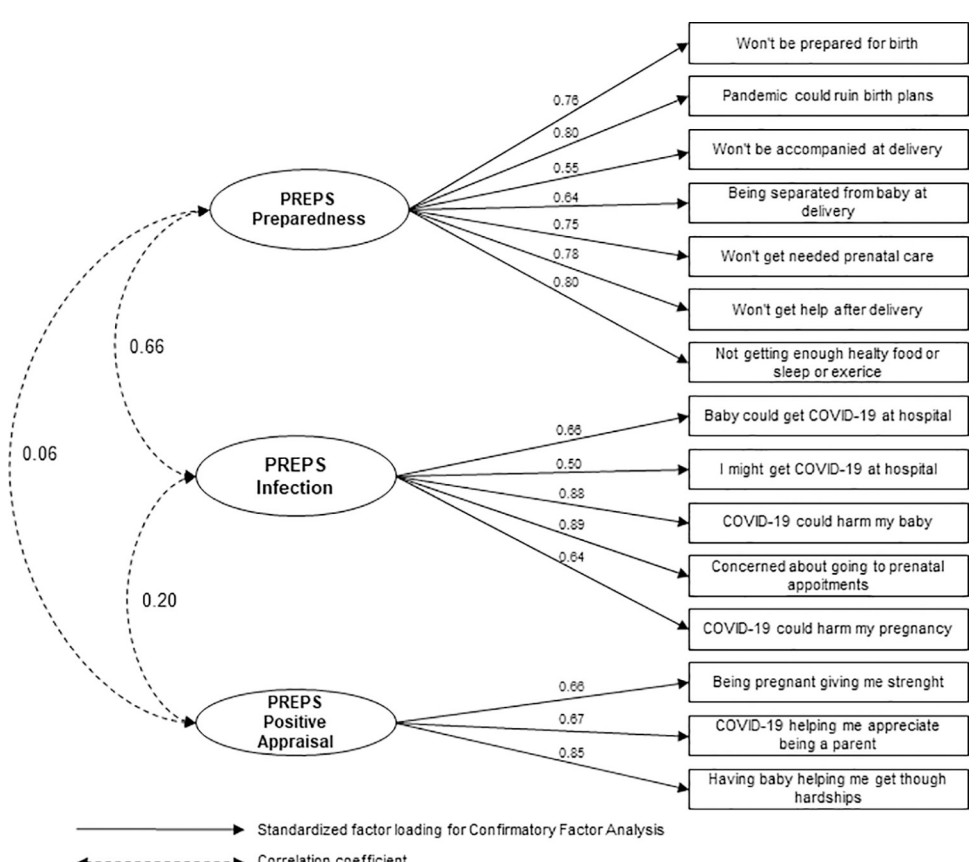

**Fig 1. Standardized estimates of confirmatory factor analysis of the Serbian translation of the Pandemic-Related Pregnancy Stress Scale.**

labeled "Preparedness Stress" which is related to stress about pregnancy and childbirth; Factor 2, labeled "Infection Stress" addresses worries about COVID-19 infection; and Factor 3, labeled "Positive Appraisal," reflects the perception of benefits associated with pregnancy during the pandemic. These results are consistent with the three-factor solution identified in the original PREPS version and also in later-validated PREPS questionnaires in other countries [14, 16, 17, 19–21]. In the Polish version, the authors suggest that the "Positive Appraisal" domain might be excluded if the focus is on stress [18]. Referring to that, Korean authors excluded that domain in their study [20].

This study showed that the Serbian translation of the questionnaire showed a very good explanatory fit to the data from the Serbian population. The variables in the dataset were sufficiently correlated for principal component analysis.

Comparative analysis of psychometric properties of the PREPS in our and other studies are shown in Table 4. The internal consistency of the total PREPS showed the highest score for the 3-factor model compared to all other analyzed studies in Spain [16], Poland [18], Greece [19] and Korea [20], noting that there were no data for total scores in studies performed in the USA [14], Germany [17] and Italy [21]. PREPS-"Preparedness Stress" and PREPS-"Infection Stress" showed good internal consistency in all other studies except the one performed in Spain, which showed acceptable consistency. The most variations were found in PREPS-"Positive Appraisal"–good internal consistency was noted in the Greek study, acceptable consistency in Serbian, German and Italian studies, questionable consistency in studies in Poland and the USA, and poor consistency in the Spanish study. Confirmatory factor analysis revealed excellent fit indices for Serbian and Polish [18] version of the PREPS, with CFI, TLI, and RMSEA values indicating strong model fit surpassing the fit indices reported for the original [14] and German [17] version of PREPS which showed marginal-good fit. Other studies [16, 19, 20, 26] showed either variable, but primarily poor fit indices, or provided partial CFA data or no data at all.

This study surveyed pregnant women by administering questionnaires during outpatient visits at community health center similar to previous research [16, 26]. In contrast, Preis et al. [14], along with Schaal et al. [17] in Germany and Ilska et al. [18] in Poland, utilized anonymous online surveys to collect data from pregnant women.

In Serbia, the average age of pregnant women in this sample, ranging from 30 to 32 years, aligns closely with findings from earlier studies [14, 16–18, 26]. In the group of Serbian women surveyed, 88.4% were either married or cohabitating, which is comparatively lower than the results from studies [17, 18], especially in studies conducted by Colli et al. in Italy (99.2%) and Garcia-Silva et al. in Spain (92.7%) [16, 26].

In our study, the employment rate among pregnant women was exceptionally high at 89.9%, surpassing the figures reported in Spain and Poland, where the employment rates were closer to 75% [16, 18].

In Serbia, the share of highly educated pregnant women was notably high. A comparable level of higher education was found in Poland, as reported by Ilska et al. [18], while Garcia-Silva et al. observed a 20% lower rate [16].

This study's frequency of primiparous women matches the results reported in Italy, Spain, Germany, and Poland, with about half of the participants being first-time mothers [14, 16–18]. Chronic disease rates among pregnant women were similar to those reported in Italy, where less than 20% of women had associated medical conditions [26].

A notable finding in this study was that the rate of couples receiving infertility treatment in Serbia was 3.2–3.6% higher compared to other studies [17, 18]. Additionally, the rate of emotional and psychiatric conditions in pregnant women in Serbia ranged from 3–4%, consistent with the results in the Italian study [26].

## Conclusion

The Serbian version of the Pandemic-Related Pregnancy Stress Scale has good psychometric properties. The PREPS-SRB questionnaire serves as a valuable tool for Serbian healthcare professionals, allowing them to identify pregnant women experiencing significant stress related to the COVID-19 pandemic.

## Supporting information

**S1 Appendix. PREPS-SRB–Translation (Serbian version).**
(PDF)

**S2 Appendix. PREPS-SRB database.**
(XLSX)

## Author Contributions

**Conceptualization:** Aleksandra Kostić, Jelena Milin-Lazović.

**Data curation:** Konstantin Kostić, Aleksandra Petrović, Andrija Vasilijević.

**Formal analysis:** Jelena Milin-Lazović.

**Investigation:** Konstantin Kostić.

**Methodology:** Konstantin Kostić, Aleksandra Kostić, Jelena Milin-Lazović.

**Project administration:** Konstantin Kostić, Aleksandra Petrović, Andrija Vasilijević.

**Supervision:** Aleksandra Kostić, Jelena Milin-Lazović.

**Writing – original draft:** Konstantin Kostić, Jelena Milin-Lazović.

**Writing – review & editing:** Konstantin Kostić, Aleksandra Kostić, Aleksandra Petrović, Andrija Vasilijević, Jelena Milin-Lazović.

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
