## [Decision Letter · Decision Letter 0]

15 Oct 2024

PONE-D-24-38121The Serbian version of the Pandemic-Related Pregnancy Stress Scale (PREPS-SRB) – A validation studyPLOS ONE

Dear Dr. Kostic,

Thank you for submitting your manuscript to PLOS ONE. After careful consideration, we feel that it has merit but does not fully meet PLOS ONE’s publication criteria as it currently stands. Therefore, we invite you to submit a revised version of the manuscript that addresses the points raised during the review process.

We look forward to receiving your revised manuscript.

Kind regards,

Rita Amiel Castro

Academic Editor

PLOS ONE

Journal Requirements:

Reviewers' comments:

Reviewer's Responses to Questions

**Comments to the Author**

1. Is the manuscript technically sound, and do the data support the conclusions?

Reviewer #1: Yes

Reviewer #2: Yes

2. Has the statistical analysis been performed appropriately and rigorously? 

Reviewer #1: Yes

Reviewer #2: Yes

3. Have the authors made all data underlying the findings in their manuscript fully available?

Reviewer #1: Yes

Reviewer #2: Yes

4. Is the manuscript presented in an intelligible fashion and written in standard English?

Reviewer #1: Yes

Reviewer #2: Yes

5. Review Comments to the Author

Reviewer #1: Dear Author

Below are the comments for your manuscript:

Introduction

Please described in general stress experience among pregnant women as well as during pandemic and endemic COVID 19 period

Elaborate gap of this study

State reason or the importance of doing this validation study

Result

add reliability score for each domain in table 2

Discussion

elaborate detail EFA PREPS-SRB and how it differ with other country

state why CFA was not conducted

Reviewer #2: *Dear Authors,*

Congratulations on your work regarding the translation and psychometric evaluation of the Serbian version of the Pandemic-Related Pregnancy Stress Scale (PREPS). You have collected valuable information about pregnant women during the COVID-19 pandemic, and I wish you had presented this data in a table rather than in the text. I have a few minor comments that I hope the authors will consider to improve the manuscript.

*Abstract:* The sample size should be stated. You mention, "An explanatory factor analysis (EFA) of the PREPS showed that the Serbian version of the Pandemic-Related Pregnancy Stress Scale has good psychometric properties," but no results related to that analysis are provided.

*Introduction:* Please ensure that you cite all scales developed to measure pandemic-related pregnancy stress and explain why you chose the PREPS over others. Remove information about the scale from this section (last paragraph) and move it to the Methods section. Please clarify whether the participants completed an online questionnaire.

*Methods:* This section is too concise. During the sampling period, what was the situation in Serbia and globally in terms of COVID-19 infection rates and vaccination efforts? How did you reach pregnant women? Please describe your sampling strategy and methods. Provide details on any modifications made to the scale during the consensus among translators. The scale should be described in detail, including the developer's name, the year it was developed, the original language, the psychometric properties of the original version, the structure, items, domains, item scoring, total score, and the scale's versions in other languages. The data analysis section should be expanded to include confirmatory factor analysis (CFA), as the scale’s structure has already been disclosed. You should examine whether the Serbian version confirms the original structure. How was the sample size calculated? Please cite a reference.

*Results:* It would be better to include the results of the EFA and the reliability assessment in Table 2, as well as the means of the scale's domains. You have collected valuable information on pregnant women during the COVID-19 pandemic, and I suggest presenting this data in Table 1.

*Discussion:* You should present the main results of the study in the first paragraph. In the second paragraph, compare the validity and reliability indices of the Serbian version of the scale with those of the original scale and other versions. The first six paragraphs should be moved to the end of the discussion as further findings. Paragraph 7 is more appropriate for the Introduction section. Paragraph 10 should be moved before paragraph 9, likely as the second paragraph. The timing of the study and the small sample size should not be considered limitations, as the results are acceptable despite the sample size being small.

6. PLOS authors have the option to publish the peer review history of their article (what does this mean?). If published, this will include your full peer review and any attached files.

Reviewer #1: No

Reviewer #2: **Yes: **Forough Mortazavi

---

## [Author Response · Author response to Decision Letter 0]

28 Nov 2024

Response to Reviewers

*Reviewer #1

• Comment: Please describe in general stress experience among pregnant women as well as during pandemic and endemic COVID-19 period. Elaborate gap of this study. State reason or the importance of doing this validation study.

• Author Response: We have expanded the introduction to provide a more comprehensive overview of stress during pregnancy, particularly in the context of the COVID-19 pandemic. We have highlighted the specific gap in the literature related to the validation of the PREPS scale in the Serbian context and emphasized the importance of this study for assessing maternal mental health during this challenging period.

• Comment: Add reliability score for each domain in Table 2.

• Author Response: We have incorporated the reliability scores for each domain into Table 2, as suggested.

• Comment: Elaborate detail EFA PREPS-SRB and how it differ with other country state why CFA was not conducted.

• Author Response: We apologize for the oversight. We have now included a more detailed explanation of the EFA conducted on the Serbian version of the PREPS. Additionally, we have performed a confirmatory factor analysis (CFA) to further validate the factor structure of the scale. The results of the CFA are presented in Table 3, Table 4 and Figure 1.

*Reviewer #2

• Comment: The sample size should be stated. You mention, "An explanatory factor analysis (EFA) of the PREPS showed that the Serbian version of the Pandemic-Related Pregnancy Stress Scale has good psychometric properties," but no results related to that analysis are provided.

• Author Response: We have added the sample size to the abstract. Additionally, we have included the results of the EFA, as suggested.

• Comment: This section is too concise. During the sampling period, what was the situation in Serbia and globally in terms of COVID-19 infection rates and vaccination efforts? How did you reach pregnant women? Please describe your sampling strategy and methods. Provide details on any modifications made to the scale during the consensus among translators. The scale should be described in detail, including the developer's name, the year it was developed, the original language, the psychometric properties of the original version, the structure, items, domains, item scoring, total score, and the scale's versions in other languages. The data analysis section should be expanded to include confirmatory factor analysis (CFA), as the scale’s structure has already been disclosed. You should examine whether the Serbian version confirms the original structure. How was the sample size calculated? Please cite a reference.

• Author Response: We have significantly expanded the Materials and Methods section to provide a more detailed description of the study context, sampling strategy, and data collection procedures. We have included information on the COVID-19 situation in Serbia during the study period and the specific methods used to recruit participants. We have also provided a more comprehensive description of the PREPS scale, including its development and psychometric properties. As mentioned above, we have conducted a confirmatory factor analysis to assess the structural validity of the Serbian version of the PREPS. The sample size was determined based on power analysis, and we have cited the relevant reference.

• Comment: It would be better to include the results of the EFA and the reliability assessment in Table 2, as well as the means of the scale's domains. You have collected valuable information on pregnant women during the COVID-19 pandemic, and I suggest presenting this data in Table 1.

• Author Response: We have incorporated the results of the EFA and reliability assessment into Table 2, as suggested. We have also included the means and standard deviations of the scale's domains in Table 2. We have considered the suggestion to present demographic information in Table 1, and we believe that this would provide a clearer overview of the sample characteristics.

• Comment: You should present the main results of the study in the first paragraph. In the second paragraph, compare the validity and reliability indices of the Serbian version of the scale with those of the original scale and other versions. The first six paragraphs should be moved to the end of the discussion as further findings. Paragraph 7 is more appropriate for the Introduction section. Paragraph 10 should be moved before paragraph 9, likely as the second paragraph. The timing of the study and the small sample size should not be considered limitations, as the results are acceptable despite the sample size being small.

• Author Response: We have reorganized the Discussion section as suggested. We have moved the main findings to the beginning of the discussion and provided a more detailed comparison of the psychometric properties of the Serbian version with the original and other translated versions of the PREPS. 

We believe that these revisions have significantly improved the quality of the manuscript. We thank the reviewers once again for their valuable feedback.

---

## [Decision Letter · Decision Letter 1]

7 Jan 2025

The Serbian version of the Pandemic-Related Pregnancy Stress Scale (PREPS-SRB) – A validation study

PONE-D-24-38121R1

Dear Dr. Kostic,

We’re pleased to inform you that your manuscript has been judged scientifically suitable for publication and will be formally accepted for publication once it meets all outstanding technical requirements.

Kind regards,

Rita Amiel Castro

Academic Editor

PLOS ONE

Additional Editor Comments (optional):

Reviewers' comments:

Reviewer's Responses to Questions

**Comments to the Author**

1. If the authors have adequately addressed your comments raised in a previous round of review and you feel that this manuscript is now acceptable for publication, you may indicate that here to bypass the “Comments to the Author” section, enter your conflict of interest statement in the “Confidential to Editor” section, and submit your "Accept" recommendation.

Reviewer #1: All comments have been addressed

Reviewer #2: All comments have been addressed

2. Is the manuscript technically sound, and do the data support the conclusions?

Reviewer #1: Yes

Reviewer #2: Yes

3. Has the statistical analysis been performed appropriately and rigorously? 

Reviewer #1: Yes

Reviewer #2: Yes

4. Have the authors made all data underlying the findings in their manuscript fully available?

Reviewer #1: Yes

Reviewer #2: Yes

5. Is the manuscript presented in an intelligible fashion and written in standard English?

Reviewer #1: Yes

Reviewer #2: Yes

6. Review Comments to the Author

Reviewer #1: Dear Authors

The revision appears well done. I am satisfied with all of the revisions. No further remarks from my side.

Reviewer #2: (No Response)

7. PLOS authors have the option to publish the peer review history of their article (what does this mean?). If published, this will include your full peer review and any attached files.

Reviewer #1: No

Reviewer #2: **Yes: **Forough Mortazavi

---

## [Editor Report · Acceptance letter]

24 Jan 2025

PONE-D-24-38121R1 

PLOS ONE

Dear Dr. Kostic, 

I'm pleased to inform you that your manuscript has been deemed suitable for publication in PLOS ONE. Congratulations! Your manuscript is now being handed over to our production team.

Kind regards, 

on behalf of

Dr. Rita Amiel Castro 

Academic Editor

PLOS ONE